# Histone Deacetylases and Their Inhibitors in Cancer Epigenetics

**DOI:** 10.3390/diseases7040057

**Published:** 2019-11-01

**Authors:** Kelly N. Hassell

**Affiliations:** Department of Biology, College of St. Elizabeth, Morristown, NJ 07960, USA; khassell@cse.edu or phluoro4@gmail.com; Tel.: +1-(215)435-4964

**Keywords:** epigenetics, cancer therapeutics, histone deacetylases, histone deacetylase inhibitors

## Abstract

Histone deacetylases (HDAC) and histone deacetylase inhibitors (HDACi) have greatly impacted the war on cancer. Their role in epigenetics has significantly altered the development of anticancer drugs used to treat the most rare, persistent forms of cancer. During transcription, HDAC and HDACi are used to regulate the genetic mutations found in cancerous cells by removing and/or preventing the removal of the acetyl group on specific histones. This activity determines the relaxed or condensed conformation of the nucleosome, changing the accessibility zones for transcription factors. These modifications lead to other biological processes for the cell, including cell cycle progression, proliferation, and differentiation. Each HDAC and HDACi class or group has a distinctive mechanism of action that can be utilized to halt the progression of cancerous cell growth. While the use of HDAC- and HDACi-derived compounds are relatively new in treatment of cancers, they have a proven efficacy when the appropriately utilized. This following manuscript highlights the mechanisms of action utilized by HDAC and HDACi in various cancer, their role in epigenetics, current drug manufacturers, and the impact predicative modeling systems have on cancer therapeutic drug discovery.

## 1. Introduction

Epigenetics is the modification of a cell’s DNA without changing its original sequence. This reprogramming can cause genes to be turned on/off depending of the intended expression for the cell. Epigenetic mechanisms allow genetically identical cells to adopt different phenotypes, regulating transcriptional availability of the genome through differential chromatin marking and packaging, in which networks of mutually reinforced or counteracting signals are created [1]. Chromatin markers can be preserved and/or changed according to environmental, developmental, or pathological needs. The genetic and epigenetic mechanisms influence each other by cooperatively enabling the initial stages of cancer cell growth [2]. The deregulation of epigenetic control in cells has been noted as a common characteristic of cancerous tumor cells. Therefore, the role of epigenetic drugs has become increasingly important in reverting the malignant phenotype. HDAC and HDACi have become more influential in epigenetics as they provide specific epi-based treatments to target specific types of cancers [1,3,4]. Acetylation neutralizes the positively charged histone lysine residue, causing a relaxed chromatin conformation which increases the accessibility of transcriptional modifiers to the gene. In the removal of an acetyl group from a histone, chromatin condensation is induced, leading to gene transcription repression [5,6].

Recently, histone deacetylases (HDAC) and histone deacetylase inhibitors (HDACi) have been effectively used to modulate acetylation in efforts to regulate the accessibility of transcription factors to DNA coding regions. HDAC and HDACi play a major role in several biological processes, such as cell cycle progression, proliferation, and differentiation. HDAC and their inhibitors can be used to post-translationally modify histones—the histone code that is read and recognized by other proteins to regulate gene expression. Unlike other histone code modifications, DNA methylation and phosphorylation effect protein interactions and expression level of HDAC and HDACi, which can be used to activate and/or repress overall gene expression [7,8]. Lysine acetylation is an example of a reversible transcriptional modification that is controlled by the antagonistic interactions found between histone acetylases (HATs) and HDAC.

## 2. HDAC Classifications

HDAC are known for their unique ability to catalyze the removal of acetyl groups from the amino-terminal lysine residues of histones and nonhistone proteins. The most commonly acetylated nonhistone protein is p53, a tumor suppressor. Mutations to p53 have been found in more than 50% of all types of cancers [9]. There are four main human classes for HDAC that have been grouped based on their homology to yeast proteins, enzymatic activity, and cellular localization.

### 2.1. Class I

Class I is found in the nucleus and includes HDAC1, 2, 3, and 8. HDAC1 and 2 are similar in their catalytic domain on the N-terminus and their interactions with other proteins in the form of a complex of which regulates their deacetylation activity. Co-Rest (Co-repressor for element-1-silencing transcription factor) is one of the known protein complexes that contain HDAC1 and 2. Co-Rest works with REST (RE-1-silencing transcription factor) to regulate neuronal cell growth [10]. The other two remaining members of class I, HDAC3 and 8, are also similar in their characterizations and can be found interacting with HDAC4, 5, and 7 within the SMRT (silence mediator for retinoid and thyroid receptors) and N-Cor (nuclear receptor co-repressor) complex formations [10]. Class I HDACs are deregulated in cancers. Their overexpression has been found in tissues from breast, gastric pathway, pancreas, lungs, and prostate. HDAC1, 2, and 3 are commonly found in renal cancer and Hodgkin’s lymphoma. The catalytic activity of HDAC1 and 2 has been used to regulate the functions of p53 [11,12]. Evidence of a frame shift gene mutation distinctively characterizes HDAC2 in class I. HDAC3 has documented interaction activity with cancer-associated genes (CAGE) in testicular cancers [13,14]. HDAC8 deacetylates lysine residues of histone and nonhistone proteins, and can be found in cervical cancers, neuroblastomas, and pancreatic ductal adenocarcinomas [15].

### 2.2. Class II

Class II HDACs are localized in the cell cytoplasm and nucleus, providing a distinctive characteristic that allows for shuttling between the two cellular compartments. There are two subclasses for Class II: HDAC4, 5, 7, and 9 are grouped as IIa, and class IIb includes HDAC6 and 10. Class IIa members: HDAC4, 5, and 7 share similarities, as well as the Class IIb members. HDAC9 contains splice variants and has a catalytic domain on its C-terminus. Evolutionary data supports the relationship of HDAC6 and HDAC10. HDAC6 contains two catalytic domains that work together in a tandem formation. However, the catalytic domain has been identified to resemble HDAC9. Data supports the anticancer properties of HDAC6; the deacetylation of microtubules in tumor cells suppresses angiogenesis, leading to the repression of metastasis. However, the expression of HDAC6 in tumor cells has been observed in patients with higher breast cancer survival rates [16,17,18]. The three nuclear export signals are significant players in the functional characteristics of HDAC6 as a catalyzed nuclear–cytoplasmic pathway with targets unrelated to transcription [19].

HDAC6 is highly expressed in breast cancer, is used as a late-stage cancer indicator, and is present in both the positive progesterone and α-estrogen receptors. Genes associated with the isoform HDAC6 are estrogen induced and are upregulated as late mRNA expressions in the cDNA microarray using MCF-7 cells. The mRNA expression levels analyzed via RT-PCR are higher in patients with small breast cancer tumors (<2 cm diameter). Survival rates in breast cancer patients can be linked to the increased HDAC6 protein and mRNA expression [18,20]. HDAC10 is the most recently characterized, with catalytic domain activity found primarily on its N-terminus and with the C-terminus consisting of a secondary catalytic domain known as the leucine-rich domain (LRD) region. The LRD region allows HDAC10 to act as a recruiter of acetyl groups in its interactions with other HDAC complexes, such as HDAC1, 2, 3, 4, 5, and 6. HDAC4 interacts with HDAC3 via N-CoR, has been found overexpressed in breast cancers, and is used as a cancer progression marker in esophageal carcinomas [21]. HDAC4 has catalytic activity that structurally regulates access to the Zn^2+^ binding domain [22]. HDAC5 and 9 are used as markers in medulloblastomas and rare form of breast cancer and can be used to determine the cancer patient’s survival [23]. HDAC5 with lysine-specific demethylase 1 proteins (LSD1) are overexpressed in estrogen receptor-negative breast cancer [24].

The regulation of HDAC5 has proven to be an effective form of therapeutic treatment. HDAC7 has been found overexpressed in pancreatic cancer and acute lymphoblastic leukemia and has been reported as insensitive to its previously designated HDACi, trichostatin A(TSA) [1,22,25]. HDAC10 is found in cervical cancer as a metastasis suppressor [26].

### 2.3. Class III

Class III is more frequently referred to as the Sirtuins (silence mating-type information regulation-2), which are required to maintain chromatin. Seven Sirtuins have been classified and found in the cytoplasm, nucleus, and mitochondria. SIRT2 is the only one found in the cytoplasm. SIRT1, 6, and 7 are found in the nucleus and SIRT3, 4, and 5 are found in the mitochondria. Sirtuins have enzymatic NAD+ (nicotinamide adenine dinucleotide)-dependent functions that regulate transcription, metabolism, and cellular stress responses. Class III (Sirtuins) has been linked to cancer pathways due to functional involvement in controlling the cell’s survival under stress conditions. SIRT1 regulates cancer cell growth and has been identified as a tumor suppressor in retinoblastoma and can be modulated in cancer by deacetylate histones [27]. SIRT1 promotes cell cycle arrest, DNA repair, and cell survival under low stress conditions. In higher stress conditions (lacking tumor suppressors and mitotic checkpoints), SIRT1 can promote tumor formation and cancerous cell growth [28,29,30,31]. SIRT2 acts as a tumor suppressor and its absence disrupts normal mitotic checkpoints at G_1_, G_2_, and metaphase, which causes increased tumorigenesis [32,33,34]. When downregulated, SIRT2 blocks cell metabolism to inhibit the metastatic spread of hepatocellular carcinoma cells [35,36,37]. SIRT3 has been found in transcription factor regulation of various types of breast cancers [38]. SIRT4 acts as a tumor suppressor in gastric cancers. It interacts with glutamate dehydrogenase and polyADP-ribose polymerase I (PARP) inhibition [39,40,41]. SIRT5 promotes cell proliferation in hepatocellular carcinoma [42,43]. However, SIRT6 plays a role in tumor suppression as it restricts the cancerous cell metabolism to protect the original genomics of the surrounding healthy cells, as observed in retinoblastoma [44]. SIRT7 has been found in the nucleolus, interacting with H3 histone, as well as in the activation of RNA polymerase I in its mechanistic action of tumoral growth inhibition in prostate and non-small cell lung cancer and osteosarcoma [45,46,47,48,49,50,51].

### 2.4. Class IV

HDAC11 has been classified by itself in Class IV and it often considered a hybrid of the HDAC’s in Class I and II. The catalytic domain of HDAC11 is in the N-terminus and resembles HDAC3 and 8, which validates its unique hybrid classification. The expression of HDAC11 has been found in the kidneys, brain, heart, testis, and skeletal muscles of the human body. Its major function has been in the association of oligodendrocyte development and immune system responses [5]. HDAC11 is overexpressed in Hodgkin’s lymphoma and interacts with HDAC1 and 2. Small interfering RNAs (siRNAs) can selectively inhibit HDAC11 expression and induce apoptosis in human leukemia cell lines and increase necrosis [52,53]. HDAC11′s lysine defatty-acylase properties are more efficient than its deacetylase activity. Probing discoveries may lead to other class IIa HDACs with similar or shared defatty-acylase ability. HDAC11 defatty-acylates substrates with an efficiency that is more than 10,000-fold greater than its deacetylase activity. This functionality of HDAC11 resembles that of the sirtuin family; catalyzing acyl group removal. Defatty-acylase activity presents as a new classification and appears evolutionarily divergent amongst the previous grouped zinc-dependent HDACs [52,54].

### 2.5. Similarities in Classes I, II & IV

HDAC Classes I, II, and IV, known as the classical HDACs, are Zn^2+^ dependent enzymes. The activities of HDACs correspond to chromatin formation, histone acetylation modulation, and the altering of transcription factors involved in the promoter regions of genes. Cancer is classified as a genetic disease that results from mutations found within the genetic code of a normal cell. Chromosomal dysfunctions occur in normal cells with regards to its ability to suppress the tumor developing genes, which causes the hyper-activation of oncogenes. The use of HDACs in targeting tumor cells is becoming the leading research method used to regulate and/or suppress the metastasis of some types of cancers. See HDAC summary Table 1 below for Classes I-IV. 

## 3. HDAC Inhibition

Just as HDAC are classified into subfamilies of multiprotein complexes, HDAC inhibitors (HDACi) have been grouped based on their ability to interfere with the function of HDAC. HDACi are cytostatic agents that modulate gene expression via indirect induction of histone acetylation. In research studies, HDACi have demonstrated interference activity within cancerous tumor cells, altering proliferation in vivo and in vitro. They act as inhibitors by inducing cell cycle arrest, differentiation, and apoptosis [55,56,57,58]. Inhibitors of HDAC may enable the re-expression of repressed regulatory genes in cancer cells and reverse their malignant phenotype. For example, sodium butyrate and trichostatin A (TSA) are known inhibitors of HDAC activity, demonstrated by the induction G_0_–G_1_ cell cycle arrest and apoptosis in the SW620 colonic carcinoma cell line [59]. HDACi inhibits the activity of HDAC enzymes, promoting the acetylation of histones and nonhistone proteins. This inhibition can increase gene expression and alter DNA processing, including replication and repair. HDACi used as epigenetic regulators in cancer therapeutics have been generated from natural and synthetic resources [60,61,62,63]. Specifically, in breast cancer cells with overexpressed hormone estrogen receptor-2 (HER2), HDACi can be epigenetically dysregulated by the phosphorylation of the transcriptional protein Sp1 motif [20,64]. They are currently classified into five different groups: hydroxamates, aliphatic acids, benzamides, tetrapeptides/depsipeptides, and sirtuin inhibitors.

### 3.1. Group 1

Group 1, the hydroxamates, consists of trichostatin A (TSA), suberoylanilide hydroxamic acid (Vorinostat), Panabinostat, Belinostat, and abexinostat hydrochloride. Of the five members in this group Vorinostat, Panobinostat, and Belinostat are most highly developed and heavily tested. Group 1, in addition to Groups 3 and 4, has been classified as inhibitors of Class I and II HDACs due to their ability to bind the Zn^2+^ ion required for HDAC enzymatic activity.

In 2006, Vorinostat was approved by the FDA for treatment of cutaneous T-cell lymphoma. Vorinostat is a second-generation polar compound that binds to the catalytic domain on HDAC. It enables the hydroxamine to form a zinc ion ring within the catalytic pockets of HDAC to inhibit its deacetylation. Vorinostat has been clinically trialed for the treatment of other forms of cancer, such as non-Hodgkin’s lymphoma, breast, and colon cancer, but did not show the same type of success as documented with cutaneous T-cell lymphomas. Vorinostat (SAHA) has been used in several clinical trials; phase III for glioblastomas, phase II for uterine sarcomas and other solid malignant tumors [65,66,67]. In the phase II clinical trial for uterine sarcoma, the study investigated the efficacy of Vorinostat as a monotherapy in patients with HDAC-positive, advanced metastatic, and mixed epithelial mesenchymal tumors after receiving antiproliferative therapy [68,69,70]. Vorinostat has been renamed Zolinza^®^ and is currently manufactured by Merck in capsule form for oral administration [65].

Panobinostat is a nonselective HDACi that has completed Phase I and II clinical trials and can be used separately or in combination with other therapeutic treatments of non-Hodgkin’s lymphoma, leukemia myeloblasts, acute myeloid leukemia, multiple myelomas, and other advanced solid tumors found in the lung and breast tissues [71,72]. When used in combination with other compounds, Panobinostat has the mechanistic ability to interfere with DNA methylation and tyrosine kinase inhibition [73]. Panobinostat keeps genes that suppress cell division and growth of cancerous cells active. In 2015, Panobinostat was approved by the FDA in combination with Bortezomib in treating patients with multiple myeloma. It has been renamed Farydak^®^ and is currently manufactured by Novartis for intravenous administration [74].

In March 2012, Belinostat was been used in various clinical trials for solid tumors and hematological cancers. Belinostat induces the accumulation of acetylated histones and proteins, thus altering the gene expression and inducing cell-cycle arrest or apoptosis of cancerous cells. When used separately, Belinostat has an antitumoral effect in the treatment of peripheral T-cell lymphoma, cutaneous T-cell lymphoma, liver cancer, and thymoma [60]. As of 2014, Belinostat was approved by the FDA for the treatment of patients with relapsed or refractory peripheral T-cell lymphoma (PTCL). It has been renamed Beleodaq^®^ and is currently manufactured by Topotarget Inc. for intravenous administration [75].

### 3.2. Group 2

Group 2 HDACi are characterized as aliphatic acids. Valproic acid (VPA) has been the most widely used and understood within group 2 HDACi. Traditionally, VPA had been used in the clinical treatment of epilepsy, bipolar disorder, schizophrenia, and in extreme cases of migraine headaches and depression. As a HDACi, it has been used in Phase I and II clinical trials to treat various types of cancers. VPA has demonstrated efficacy when used in combination with other anticancer compounds that are known for therapeutic treatment of lymphocytic leukemia, acute myeloid leukemia, melanoma, HIV infections, and autoimmune lymphoproliferative syndrome. The short-chained fatty acid structure of VPA enhances the mechanism of action in treating glioblastoma and breast cancer patients [67,76,77,78,79].

### 3.3. Group 3 

In Group 3, Entinostat is a synthetic derivative that inhibits Class I and II HDACs. Entinostat is a selective autophagy inducer, an orally administered drug that is in development by Syndax Pharmaceuticals with exemestane for advanced hormone receptor breast cancer [80]. Clinical research supports the activity of Entinostat as an antitumor promoter and inhibitor of HDAC activity. While it has not been approved by the FDA for monotherapeutic use, it has been approved in combination with anticancer tumor compounds [60,80,81,82].

### 3.4. Group 4 

Group 4, characterized by tetrapeptide structure (also called depsipeptides), consists of Apicidin and Romidepsin. This group is also known as the bicyclic peptides that inhibit Class I and II HDACs. Apicidin and Romidepsin are clinically proven to have potent cytotoxicity against malignant cancer cells in vivo and in vitro. In patients with colorectal, renal, and breast cancers, depsipeptides have been administered with short-term toxicity. For example, Romidepsin (Istodax^®^) is a naturally harvested product from bacteria and is currently being used in cancer treatments due to its toxicity observed in the treatment of peripheral T-cell lymphoma [83,84]. The mechanism of action utilized by Romidepsin triggers the accumulation of acetylated histones to induce apoptosis in cancer cells [85,86]. Since 2009, Romidepsin has been successfully included in over 50 interventional drug trials. As of 2011, the FDA approval for Istodax^®^ incorporated usage in the therapeutic treatment of patients with peripheral T-cell lymphoma. Currently, Istodax^®^ is manufactured by Celgene Corporation as injectables [83].

### 3.5. Group 5 

The sirtuin inhibitors of Group 5 include physiological inhibitors—nicotinamide, cambinol, and sirtinol derivatives—that work specifically on SIRT1 and 2. Of the seven human sirtuins, SIRT1 and 2 are upregulated in cancerous tumors and are therefore the most studied. Since SIRT1 and 2 have the ability to inactivate proteins like p53 in transcription and post-translation, developing inhibitors could provide a wealth of anticancer agents used for various treatments. For example, benzimidazole modulates antiproliferation cell activity and has proven efficacy in two different types of breast cancer cell lines; luminal and basal A subtype [87]. The specific mechanism used by benzimidazole to inhibit SIRT1 and 2 has not been clearly defined, but these novel derivatives provide a new trajectory for developing new therapeutic drug agents to fight various types of cancers. Another inhibitor of SIRT1 and 2 activities is cambinol and its derivatives. In vitro and in vivo research studies demonstrate cambinol’s antilymphoma abilities. Cambinol inhibits SIRT1 and 2 by inducing the hyperacetylation of p53 [88]. Unfortunately, there are two negative outcomes in using cambinol for therapeutic treatments: its moderate level of potency and poor stability. Despite the minor drawbacks, cambinol is still a promising anticancer drug agent and has a good foundation for large scale optimization [27,29,31,89]. See HDACi summary Table 2. 

## 4. Advances in Studying HDAC/HDACi in Cancer

The research and development of cancer therapeutics that was once limited to time consuming wet-lab procedures to has evolved to incorporate complex high throughput bioinformatics analysis. Several research groups globally have made significant contributions to our understanding of the mechanistic influence of HDAC/HDACi in various types of cancers. Researchers that have employed the use of in silico techniques utilize vast genomic databases in concert with or in lieu of wet-lab bench analysis and have revolutionized our acquisition of knowledge and understanding. In studying HDAC/HDACi compounds, utilizing the configuration of their Zn^2+^ binding domains (ZBD) further enriches our capacity to understand all possible functional interactions for designing specific drug targets. Zinc-binding domains are the key to unlock the targeted active sites within HDAC structures that can be utilized by the HDACi groups to execute inhibition. Zinc ions and hydroxamic acid have a high affinity; both are essential to ZBD structural design in HDACs. The combination of the ligand–receptor interaction and the cap end group with the presenting residues located at the HDAC active sites are the agents of specificity for HDACi [90]. The intricacies of the cap end to linker residues in ZBD were characterized most efficiently via in silico approaches. Hsu et al. designed an amino acid sequence to use in their homology modeling of HDAC5 and 9. With access to GenBank and the SWISS-MODEL server, targeted template alignment analysis generated 3D predictive models for HDAC interactions. In simulations, protein interactions were visualized in BIOVIA DS and PyMOL software [91,92,93]. In one screening process, the Hsu group identified six novel nonhydroxamate inhibitors for specifically targeting class IIa HDACs. The use of bioinformatics allowed this one group to make a huge leap in their understanding of how to target the mechanism of action via localized active sites and conformational models.

## 5. Conclusions

Research studies of HDAC and HDACi have led to major advances in the therapeutic treatment of various forms of cancer. There are pitfalls associated with using traditional research methods, specifically time and processing requirements. When using computer simulations, human error can still exist. However, the impact of vast database analysis by supercomputers and artificial intelligence is mandatory in drug discovery. In silico approaches have proven to be resourceful in gaining knowledge of cancer epigenetics. The software and hardware abilities of supercomputers can physically outperform any lab scientist. The next generation of researchers will need to have the training and knowledge necessary to program supercomputers of tomorrow to keep up with the large datasets processed daily for therapeutics. Conceptualizing the many different mechanistic actions between HDAC and HDACi has altered the development of targeted cancer therapeutics. The sirtuins and their inhibitors appear to be the most promising in terms of pharmaceutical marketability due to their duel functionality in the regulation of the cell cycle, apoptosis, and deacetylation of p53, respectively. However, when taking into account the mechanisms that are used in manipulating a cell’s genetic code, there could be several other keys to be found that can unlock the epigenetic codes of cancer. In silico approaches are a great way to explore a path of thought without enduring laborious wet bench procedures—a lifetime saving approach.

## Figures and Tables

**Table 1 diseases-07-00057-t001:** HDAC Summary.

Class:	Localization:	HDAC	Characteristics:	Activity in Cancer:	Refs.
I*	nucleus	1	N-terminus catalytic domain	Overexpressed in tissues from breast, gastric, pancreas, lungs, cervical and prostate cancers	[10,11,12]
		2			[10,11,12]
		3	Interaction with HDAC4, 5, 7 & cancer-associated genes (CAGE)		[13,14]
		8	Interaction with HDAC4, 5, 7		[15]
IIa*	cytoplasm & nucleus	4	C-terminus catalytic domain combines with HDAC3 via N-CoR, catalytic activity structurally regulates access to the Zn^2+^ binding domain	Suppresses p21; overexpressed in breast, colon, ovarian and gastric cancers	[22]
		5	Interacts with lysine-specific demethylase 1	Markers in medulloblastomas & breast cancer	[24]
		7	C-terminus activity; non-deacetylase dependent	Overexpression in pancreatic cancer & acute lymphoblastic leukemia	[11,22,25]
		9	Splice variants with catalytic domain on its C-terminus	Markers in medulloblastomas	[25]
IIb*	cytoplasm & nucleus	6	2 tandem catalytic domains	Highly expressed in breast cancer; stage indicator	[16,20]
	cytoplasm	10	Catalytic domain activity at N-terminus & C-terminus leucine rich domain (LRD)	Cervical cancer as a metastasis suppressor	[21]
III**	nucleus	SIRT1	Lys382 residue deacetylate of H1, H3, and H4; C-terminus p53 acetylation regulator	Tumor suppressor in retinoblastoma	[27,28,29,30,31]
		SIRT6	Glycolysis regulator in cancer cells	Tumor suppressor in retinoblastoma	
		SIRT7	Deacetylates lysine 18 residue of H3; succinyls activity	Ovarian, colorectal, osteosarcoma, prostate, hepatocellular, breast & non-small cell lung	[45,46,47,48,49,50,51]
	cytoplasm	SIRT2	Deacetylating α-tubulin	Tumor suppressor; ovarian, breast, leukemia, neuroblastoma, pancreatic & hepatocellular	[32,33,34,35,36,37]
	mitochondria	SIRT3	Transcription factor regulation via deacetylation	Transcription factor regulation in breast cancer	[38]
		SIRT4	Glutamate dehydrogenase and poly ADP-ribose polymerase (PARP) inhibition	Tumor suppressor in gastric cancers	[39,40,41]
		SIRT5	Promotes cell proliferation	Hepatocellular carcinoma	[44]
IV*	nucleus	11	Interacts with HDAC1 & 2; defatty-acylate substrate activity	Overexpressed in Hodgkin’s lymphoma	[5,52,53,54]

* Zn^2+^ dependent; ** NAD+ dependent.

**Table 2 diseases-07-00057-t002:** HDACi Summary.

Class:	HDACi	Characteristics:	Activity in Cancer:	Drug Name:	Refs.
I	suberoylanilide hydroxamic acid (Vorinostat)	Inhibits HDAC class I & II via Zn^2+^ ion interactions	Induces apoptosis in T-cell lymphomas, thymoma & liver	Zolinza	[65,66,67,68,69,70]
	Panobinostat			Farydak	[71,72,73,74]
	Belinostat			Beleodaq	[60,75]
	valproic acis (VPA)			N/A	[67,76,77,78,79]
II	Entinostat	Synergistic enhancement obsevered with other anticancer compounds due to short-chained fatty acid	Lymphocytic leukemia, acute myeloid leukemia, melanoma & glioblastoma	N/A	[60,80,81,82]
III	Apicidin	Benzamide group; inhibits HDAC class I & II; selective autophagy inducer	Antitumor promoter in hormone receptor breast cancer	N/A	[83]
IV	Romidepsin	Bicyclic peptides; inhibit HDAC class I & II; triggers the accumulation of acetylated histones to induce apoptosis in cancer cells	Colorectal, renal, breast cancers & T-cell lymphoma	Istodax	[83,84,85,86]
V	Cambinol	Inhibits SIRT1 and 2 by induced hyperacetylation of p53	Inhibits SIRT1 and 2 by inducing the hyperacetylation of p53	N/A	[29,31,27,88,89]

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
