# Peer review of "Histone Deacetylases and Their Inhibitors in Cancer Epigenetics"

_diseases, 2019, doi:10.3390/diseases7040057_

Round 1

Reviewer 1 Report

This is a review articles on HDAC biology and its inhibitors.  Overall the article is well written but there are some latest development that are not included in the article.

1) HDAC11 biology has significant development recently, HDAC11 has been identified to be deacylase for long-chain fatty acid

Proc Natl Acad Sci U S A. 2019 Mar 19;116(12):5487-5492. doi: 10.1073/pnas.1815365116.

Cell Chem Biol. 2018 Jul 19;25(7):849-856.e8. doi: 10.1016/j.chembiol.2018.04.007. Epub 2018 May 3.

2) HDAC6 domains have different roles

Nat Chem Biol. 2016 Sep;12(9):741-7. doi: 10.1038/nchembio.2134. Epub 2016 Jul 25.

3) Hydrazide HDAC inhibitors are the latest class of HDACi

Chem Biol. 2015 Feb 19;22(2):273-84. doi: 10.1016/j.chembiol.2014.12.015.

J Med Chem. 2016 Nov 10;59(21):9942-9959. Epub 2016 Oct 26.

I believe that iincluding these aspects into the review will significantly improve the article's novelty.

Author Response

Thank you for your time in reviewing my manuscript. I have added to the referenced material as per your request.  

Reviewer 2 Report

The present review aims to describe recent research studies of both HDAC and HDACi in the therapeutic treatments in various cancers.

The review describes each HDAC class and their inhibitors but do not speculate on the latest knowledge in cancer treatments and lacks of criticism. The manuscript needs an extensive revision and English editing. There are already many reviews that describe the role of HDACs and HDAC inhibitors in cancer and the author should refer to very recent literature in order to provide useful information to the reader. Moreover, the review is not structured with figures/tables, which could be useful to summarize possible mechanisms of action, recent literature data, cellular signalling, pathways involved, preclinical and clinical studies and so on.

Author Response

Thank you for your time in reviewing this manuscript. I'd added my critique, but could not add my table due to template issues. please see attachment. 

Reviewer 3 Report

In the manuscript named "Histone Deacetylases and their Inhibitors in Cancer Epigenetics," of the author Hassell, Kelly N., is presenting the classification of histone deacetylases (HDAC) and currently commercially available HDAC inhibitors (HDACi).

After reading the manuscript, I have some major issues with this review.

I am not sure about the goal and what is the author trying to achieve. The Review, as it is, lacks a novelty, only repeats the well-known facts that can be found by a quick Google search. Not to mention, that it is too short and contains only 26 citations.

Considering the fact that there no limitation for length of Reviews in Diseases Journal, I would like to suggest two approaches for the author on how to rewrite the manuscript:

1) to write an extended version, to go deeper into the topic, and to support the general claims by the description of experiments or by data from clinical trials. As an inspiring example could serve the review from T. Kouzarides (2007): Chromatin Modifications and Their Function.

2) or to write the shorter, but more focused review with deep insight into HDAC and HDACi. For example, by selecting only one class of HDAC/ HDACi.

Also, I would like to encourage the author to use Figures or tables to make the manuscript more attractive to the readers as well as to used more than 26 citations (based on experience at least 100 citations). As a good example could serve a recently published paper (found at the Diseases Journal webpage) by Jarrell et al. (2019): Epigenetics and Mechanobiology in Heart Development and Congenital Heart Disease.

The manuscript is not suitable for being published in the present form.

Author Response

Thanking you in advance for your time and helpful comments in reviewing this manuscript. I've generated a table but the document template is preventing it from being added to the document. please see attachment.

Round 2

Reviewer 3 Report

The manuscript named "Histone Deacetylases and their Inhibitors in Cancer Epigenetics," of the author Hassell, Kelly N., has undergone significant changes. The author extended the text and implemented the reviewer's suggestions nicely.
To the rewritten version of the text, I have only minor comments:

Overall, it is hard to follow the changes in the manuscript, since the interactive and flexible .docx format has been converted into the .pdf.
Row 54-55: there is no need to re-introduce the abbreviations HDAC and HDACi since they have been mentioned before in the abstract as well as in the paragraph above (row 47)
Row 77: it should be Class I (Roman Numeral) instead of Class 1.
Row 81: the full name of the Co-Rest factor should be mentioned in the previous sentence (row 80), where the factor is firstly described.
Row 107-108: the incomplete first sentence: "...between the two ???".
Row 135: misspell HDAD4, instead of HDAC4.
Row 184: the author should clarify what the abbreviation PARP stands for and which members of the PARP family of proteins are interacting with SIRT4.
Row 255: Table 1. HDAC Summary, the HDAC Classes, should be in Roman Numerals, as well I had not found the explanations of the meaning of asterisks in Table 1.
Row 423-426: "Researchers that have...years to synthesize." is a long and hard to read the sentence, should be rewritten and shortened.

In this stage, I can recommend the acceptance of the manuscript for publishing.

Author Response

Dear Reviewer 3,

Thank you for your time and constructive comments. The line by line notations were much appreciated. I have completed your following requests:

Row 54-55: there is no need to re-introduce the abbreviations HDAC and HDACi since they have been mentioned before in the abstract as well as in the paragraph above (row 47)

Row 77: it should be Class I (Roman Numeral) instead of Class 1.

Row 81: the full name of the Co-Rest factor should be mentioned in the previous sentence (row 80), where the factor is firstly described.

Row 107-108: the incomplete first sentence: "...between the two ???".

Row 135: misspell HDAD4, instead of HDAC4.

Row 184: the author should clarify what the abbreviation PARP stands for and which members of the PARP family of proteins are interacting with SIRT4.

Row 255: Table 1. HDAC Summary, the HDAC Classes, should be in Roman Numerals, as well I had not found the explanations of the meaning of asterisks in Table 1.

Row 423-426: "Researchers that have...years to synthesize." is a long and hard to read the sentence, should be rewritten and shortened.

Sincerely, 

Kelly N Hassell